# Reliability and Validity of the Brief Emotional Experience Scale (BEES) as a Measure of Emotional Well-Being

**DOI:** 10.3390/bs15050643

**Published:** 2025-05-09

**Authors:** Shane L. Rogers, Nicole Brown, Matthew Goulding, Kathryn Campbell, Brennen Mills, Ross Hollett, Travis Cruickshank, Kazunori Nosaka

**Affiliations:** 1School of Arts and Humanities (Psychology), Edith Cowan University, Joondalup, WA 6027, Australia; r.hollett@ecu.edu.au; 2Association of Independent Schools of Western Australia, Perth, WA 6017, Australia; 3Institute of Education and Humanities, University of Wales Trinity Saint David, Lampeter SA48 7ED, UK; 4School of Medical and Health Sciences, Edith Cowan University, Joondalup, WA 6027, Australia; b.mills@ecu.edu.au (B.M.); t.cruickshank@ecu.edu.au (T.C.); k.nosaka@ecu.edu.au (K.N.)

**Keywords:** brief emotional experience scale, reliability, convergent validity, emotional well-being, factor analysis, adjective-based questionnaire, self-report measurement, psychometric evaluation, emotion measurement

## Abstract

This study presents initial reliability and validity evidence for the Brief Emotional Experience Scale (BEES) as a measure of emotional well-being. Using ordinal confirmatory factor analysis across three cross-sectional samples, Australian university students (*n* = 1239), the general public (*n* = 5631), and school students from Australia and the UK (*n* = 767). A correlated two-factor structure was supported. In the university sample, the BEES demonstrated strong convergent validity with other well-being measures and was linked to the lowest levels of reported distress when completing the survey. Preliminary cut-offs for high emotional distress were developed via comparison with the Kessler Psychological Distress Scale (K10), identifying around 20% of females and 10% of males as highly distressed. The findings of this research indicate the BEES can be utilised as a simple, flexible, and low-burden measure of emotional well-being.

## 1. Introduction

The Brief Emotional Experience Scale (BEES) is a short, adjective-based self-report tool for assessing emotional well-being and emotional experience. Previous studies employing the BEES have demonstrated good internal consistency and significant correlations with various indicators of well-being, including physical health, sense of belonging, quality of social relationships, and concerns regarding body weight and shape ([43]; [46]; [47]; [48]). Despite these promising initial findings, important questions about the BEES remain unanswered, particularly concerning its underlying factor structure, convergent validity with established emotional well-being measures, interpretability through categorical scoring bands, and participant comfort when completing the measure. This study seeks to fill these knowledge gaps by systematically evaluating the reliability and validity of the BEES across multiple samples.

Measuring emotional well-being is crucial in psychological research and practice, as it captures individuals’ general affective states, reflecting the frequency and intensity of positive and negative emotions ([9]; [13]; [38]; [63]). Emotional well-being is commonly distinguished from emotional experience, which refers to immediate affective responses to specific events. For example, emotional well-being can be assessed by asking “How have you generally felt this past week?”, whereas emotional experience would involve questions like “How did you feel during your recent exam?”. Although related, emotional well-being captures a broader, more stable affective perspective, whereas emotional experience captures context-specific emotional responses. Importantly, the BEES is specifically designed as a versatile measure capable of assessing both general emotional well-being and situation-specific emotional experiences.

Existing measures of emotional well-being include both statement-based and adjective-based scales. The widely used Depression, Anxiety and Stress Scales (DASS-21) ([19]; [32]) exemplifies a statement-based approach, assessing negative emotional states in clinical contexts. Conversely, the Positive and Negative Affect Schedule (PANAS) ([57]), an adjective-based measure, assesses both positive and negative affect. However, the PANAS emphasizes high-arousal emotions such as excitement or fear, omitting low-arousal and core valence-based emotions like happiness or sadness ([57]; [58]). This limitation has led to conceptual confusion, with the PANAS often misinterpreted as measuring emotional valence rather than arousal ([30]).

In response to these conceptual limitations, Diener and colleagues developed the Scale of Positive and Negative Experience (SPANE) ([14]), designed explicitly to measure emotional valence. The SPANE uses simple adjectives representing pleasant and unpleasant emotions (e.g., happy, sad) to clearly capture the valence dimension of affect. Although the SPANE addressed key limitations of the PANAS, the need for a shorter and more flexible measure remained.

To address this need, the BEES was developed, inspired primarily by the SPANE but with deliberate structural innovation. Unlike existing adjective-based scales, the BEES employs intentionally paired positive and negative adjectives (e.g., happy–sad, calm–worried, confident–afraid) to ensure conceptual and structural balance across positive and negative emotions ([42]). These adjective pairs were carefully selected from broader emotional categories identified by [10] ([10]), focusing on core emotional adjectives that are broadly understood and easily accessible across diverse populations. This design allows the BEES to measure emotional well-being concisely, flexibly, and intuitively, making it suitable for individuals across a wide age range and varying literacy levels ([42]).

Despite the practical use of the BEES in several studies ([40]; [42]; [43]; [46]; [47]; [48]; [61]), systematic psychometric evaluation has been limited. One recent Turkish adaptation ([7]) provided preliminary evidence of a correlated two-factor structure (positive and negative emotions), but this work was restricted to a single cultural context and did not explore alternate structural models or scoring methods.

The present study expands on these previous efforts by evaluating the psychometric properties of the BEES across three large English-speaking samples (university students, general public, and school students). We examine both the originally hypothesized correlated two factor model (positive and negative dimensions) and an alternative single-factor model to identify the optimal factor structure. Additionally, we investigate convergent validity by corelating BEES scores with established emotional well-being measures, specifically the DASS-21, PANAS and the Kessler Psychological Distress Scale (K10). We selected the PANAS as a comparator due to its prominence in the literature, shared adjective-based format, and historical conceptual confusion regarding valence and arousal which are issues the BEES explicitly seeks to clarify and resolve. We hypothesized that the BEES would show strong negative correlations with these negatively framed emotional well-being measures, providing support for convergent validity.

Furthermore, we evaluate participants’ self-reported discomfort associated when completing the BEES relative to other measures, addressing an important yet commonly overlooked aspect of survey acceptability. Based on prior research examining participant experiences with emotional well-being questionnaires ([12]; [21]; [28]; [49]; [54]; [59]), we hypothesized that discomfort would be relatively low across all measures included in the study. Comparisons between the BEES and other well-being measures formed an exploratory component of this investigation.

A final aim of the study was to develop a categorical scoring scheme (i.e., score bands) for the BEES, offering an alternative approach to reporting and interpreting results to enhance interpretability. We develop scoring bands for the BEES using established thresholds from the widely used K10 as a reference. By addressing these aims, the current study seeks to provide comprehensive psychometric evidence supporting the BEES as a valid, reliable, and accessible measure of emotional well-being.

## 2. Materials and Methods

### 2.1. Participants

Data from three distinct samples are reported in this study:University students: A sample of 1239 psychology students from a university in Western Australia (84% female; *M* age = 29.2 years, *SD* = 10.35; age range = 18–71). Ethics approval: 17563.General public: A community sample of 5631 Australian adults (57% female; *M* age = 49.6 years, *SD* = 17.2; age range = 18–100). Ethics approval: 01858. While this sample reflects broad diversity in age, education, employment, and household income, stratified sampling was not used to match the Australian population profile. Further demographic details are available in the online Appendix A (general public sample demographic information).School students: A total of 767 students aged 11–16 years were recruited from two schools: one in Australia (*n* = 354; 54% female) and one in the United Kingdom (*n* = 413; 45% female). Participants were approximately evenly distributed across year levels: ages 11–12 (24%), 12–13 (24%), 14–15 (26%), and 15–16 (26%). Ethics approval: 03410.

All ethics applications were reviewed and approved by the university ethics committee. All procedures were approved by the Edith Cowan University Human Research Ethics Committee. For the university and general public samples, informed consent was implied through voluntary completion of the anonymous online survey. For the school student sample, an opt-out consent procedure was used as part of a broader mental health screening initiative within the schools. This approach has been recommended in school-based mental health research to ensure that students most in need of support are not systematically excluded ([6]; [8]; [45]; [50]).

### 2.2. Measures

The Brief Emotional Experience Scale (BEES). All three samples completed the Brief Emotional Experience Scale (BEES) ([42]; [43]), a self-report measure consisting of six adjectives: three positive (happy, calm, confident) and three negative (worried, sad, afraid). Participants rate each adjective on a 4-point scale: (0) Not at all, (1) A little bit, (2) Quite a bit, (3) A lot. Positive and negative item scores are summed separately, and the negative score is subtracted from the positive score to produce an overall emotional well-being score ranging from −9 to +9. The two adult samples (university and general public) were instructed to reflect on their emotional experiences over the past month. In contrast, the school student sample reflected on the past week. This shorter timeframe was used because the BEES was embedded within an ongoing project exploring its use as a weekly emotional well-being monitoring tool for young people.

In addition to the BEES, university student participants completed several self-report measures assessing emotional well-being and distress. All measures were completed with reference to the past month. After each questionnaire, participants rated their level of discomfort using a 4-point scale: (0) No discomfort, (1) Low discomfort, (2) Moderate discomfort, (3) High discomfort. Time taken to complete each questionnaire was recorded, and the order of presentation was randomised across participants. All university student participants completed all measures.

The Depression, Anxiety, and Stress Scales 21-item version (DASS-21). The DASS-21 ([32]) includes 21 items assessing symptoms of depression, anxiety, and stress. Each subscale contains 7 items (e.g., depression: “I couldn’t seem to experience any positive feeling at all”; anxiety: “I felt scared without any good reason”; stress “I found it hard to wind down”). Items are rated on a 4-point scale: (0) Did not apply to me at all, (1) Applied to me to some degree, or some of the time, (2) Applied to me to a considerable degree or a good part of the time, (3) Applied to me very much or most of the time. While the DASS-21 is structured into three subscales, these are often highly intercorrelated, and the utility of reporting them separately has been questioned ([37]; [62]). For the current study, we used an overall emotional distress score by summing all 21 items, yielding a possible range of 0–63.

Positive and Negative Affect Schedule (PANAS). The PANAS ([57]) includes 20 emotion-related adjectives, 10 positive (e.g., Interested, Excited, Strong) and 10 negative (e.g., Distressed, Upset, Guilty). Participants rate the extent to which they have felt each emotion using a 5-point scale: (1) Very slightly, or not at all, (2) A little, (3) Moderately, (4) Quite a bit, (5) Extremely. Positive and negative scores are calculated by summing relevant items, with each subscale ranging from 10 to 50. Scores are summed separately to create separate overall positive and negative scores ranging from 10–50. Unlike the BEES, the PANAS was not designed to provide a balanced composite score across valence dimensions, so separate subscale scores are typically reported ([14]).

Kessler Psychological Distress Scale (K10). The K10 ([24]) consists of items measuring general emotional distress (e.g., “About how often do you feel hopeless?”). Items are rated on a 5-point scale: (1) None of the time, (2) A little of the time, (3) Some of the time, (4) Most of the time, (5) All of the time. All items are summed to yield a total score ranging from 10 to 50.

## 3. Results

### 3.1. Factor Analysis of the BEES Across Three Samples

Prior to conducting factor analyses we checked the assumptions that there would at least be consistent moderate inter-correlations among the BEES individual items. This assumption was confirmed across all samples, see Table 1.

We conducted a confirmatory factor analysis (CFA) using Mplus to test both a one-factor and a correlated two-factor model of the BEES across all samples. In the two-factor model, items were grouped by valence, with positive and negative adjectives loading onto separate but correlated factors. This structure mirrors that of the SPANE, which the BEES was largely modelled on. The CFA was estimated using the weighted least squares mean and variance-adjusted estimator (WLMSV), appropriate for ordinal data ([15]; [26]). Model fit was evaluated using [3]’s ([3]) recommended guidelines: CFI ≥ 0.95, SRMR ≤ 0.08, RMSEA ≤ 0.05.

As shown in Figure 1, the correlated two-factor model demonstrated superior fit across all samples. Factor correlations between the positive and negative emotion factors ranged from −0.67 to −0.72, indicating strong negative associations. These findings are consistent with [14] ([14]), who, in comparing the SPANE with the PANAS, concluded that the SPANE is “more saturated with the valence dimension of the emotion circumplex” (p. 5). A similar explanation applies here, supporting the strong association between positive and negative emotion factors within the BEES. To further examine the structural stability of this two-factor model, we conducted separate CFAs by sex and by age group. The model fit indices and factor loadings remained consistent across these subgroups, providing preliminary support for measurement invariance. Full results are available in the online Appendix A.

The CFA results provide strong evidence for the internal structure and consistency of the BEES. As with the SPANE, the BEES can be used to generate separate positive and negative scores. However, given the high correlation between these dimensions, it is also justifiable to use a single overall emotional well-being score, calculated by subtracting the negative from the positive score. Table 2 presents overall descriptive statistics for the three study samples.

### 3.2. Convergent Validity Between the BEES and Other Measures of Psychological Well-Being in the University Student Sample

Another aim of this study was to examine associations between the BEES and other self-report measures of emotional well-being to assess convergent validity. The university student sample completed three established instruments alongside the BEES: the DASS-21, PANAS, and K10. Descriptive statistics for all measures are provided in Table 3. The BEES demonstrated a very short completion time, with a median duration of 26.37 s.

As shown in Table 4, the BEES demonstrated strong correlations with all other emotional well-being measures. These associations provide robust evidence for the convergent validity of the BEES. Importantly, comparable patterns were observed when analyses were repeated separately by sex and age group, indicating that convergent validity was consistent across demographic subgroups (see online Appendix A).

### 3.3. Discomfort Associated with Completing Psychological Well-Being Questionnaires in the University Student Sample

Another aim of this study was to examine participant self-reported discomfort when completing each questionnaire. After responding to each measure, university student participants rated how uncomfortable they felt while completing it.

As shown in Figure 2, most participants reported no discomfort, ranging from 53% to 76% across the different measures. Only a small proportion reported moderate to high discomfort, ranging from 4% to 15%. Among all instruments, the BEES was associated with the lowest reported discomfort. A Friedman ANOVA indicated a significant difference in discomfort ratings across measures, χ^2^(3) = 401, *p* < 0.001. Pairwise comparisons revealed that discomfort associated with the BEES was significantly lower than that of the other measures (all *ps* < 0.001).

### 3.4. Cut-Off Values Determined via Examining BEES Distributions Across K10 Categories

A final aim of the research was to provide a categorical scoring system (i.e., score bands) for the BEES. To inform this, we examined the distribution of BEES scores in the university student sample, stratified by Kessler Psychological Distress Scale (K10) categories that reflect varying levels of psychological distress ([1]). These categories use plain-English descriptions with hedging language (e.g., “may”) to acknowledge the uncertainty inherent in screening measures:The individual may currently not be experiencing significant feelings of distress (K10 = 10–15).The individual may be experiencing moderate symptoms of depression and/or anxiety (K10 = 16–30).The individual may be experiencing some form of depression and/or anxiety (K10 = 31–50).

Aligning BEES scores with these K10 categories allowed for the estimation of preliminary cut-off values. As shown in Figure 3, the 25th–75th percentile range (interquartile range) of BEES scores for each K10 group showed a clear pattern of separation, supporting the utility of the BEES for broad classification of psychological distress.

Based on this pattern, we propose using the interquartile boundaries of the K10 moderate group to guide the BEES scoring bands:BEES > +3: Low likelihood of significant distress.BEES between +3 and −1: Moderate likelihood of distress.BEES < −1: High likelihood of distress.

For parsimony, we refer to these categories as low, moderate and high levels of psychological distress, respectively. To illustrate how this scoring scheme functions across different populations, Figure 4 shows the proportion of participants in each BEES category across all three samples: university students, general public, and school students.

To further evaluate the ability of the BEES scores to discriminate between individuals with high psychological distress, ROC and precision-recall analyses were conducted using K10 scores as the reference criterion. As higher BEES scores reflect greater emotional well-being (i.e., lower distress), BEES scores were inverted prior to analysis. The ROC analysis yielded an area under the curve (AUC) of 0.89, 95% CI [0.87. 0.91], *p* < 0.001, indicating excellent classification performance, see Figure 5.

Classification accuracy was evaluated at multiple thresholds, see Table 5. The cut-off of BEES < −1 yielded 71.6% sensitivity, 87.2% specificity, 56.3% precision (Youden’s index = 0.59). While BEES < 0 had a slightly higher Youden’s index (0.61) with greater sensitivity (81.9%), this came at the cost of lower specificity (79.2%) and precision (47.6%). To reduce false positives in a screening context ([18]; [39]; [64]), BEES < −1 was retained as the recommended threshold for identifying individuals at risk of high psychological distress.

## 4. Discussion

The primary aim of this study was to provide further evidence for the reliability and validity of the Brief Emotional Experience Scale (BEES) as a measure of emotional well-being, building on findings from earlier research ([7]; [42]; [43]; [46]; [47]; [48]). Across three large and diverse samples, university students (*n* = 1239), general public (*n* = 5631), and school students (*n* = 767), the BEES demonstrated good internal consistency, with McDonald’s omega coefficients ranging from 0.76 to 0.85.

Confirmatory factor analysis on the BEES supported a correlated two-factor model comprising positive and negative emotions, consistent with the SPANE ([14]). The strong inverse correlations between these factors (ranging from −0.67 to −0.72) reinforce the conceptualisation of emotional well-being as a balance between affective valences, where increases in positive emotion typically correspond with decreases in negative emotion ([9]; [14]; [22]; [23]; [38]; [41]; [63]). This pattern supports the calculation of separate positive and negative scores, while also justifying the practical use of a single overall BEES score to represent balanced emotional well-being.

This finding contrasts with the PANAS ([57]; [58]), which measures largely independent dimensions of high-arousal positive and negative affect. Its emphasis on activation, rather than valence, has been criticised for limiting its relevance to broader well-being assessment ([4]; [14]; [17]; [30]). In contrast, the BEES was developed to better reflect valence by including balanced positive and emotion pairs (e.g., happy–sad, calm–worried, confident–afraid). While such pairs are not strict opposites and can co-occur in complex emotional states (e.g., bittersweet, ambivalence, relief) ([27]; [31]; [35]), their inclusion reflects a deliberate effort to cover a representative spread of affective experience. The resulting correlated two-factor structure aligns with this design intent and supports the use of the BEES within frameworks that conceptualise emotional well-being in terms of valence balance.

In the university student sample, the BEES demonstrated good convergent validity, showing strong correlations with the K10, DASS-21, and PANAS. These results align with typical inter-scale correlations reported in the emotional well-being literature ([14]; [23]). Although the SPANE, the measure most structurally like the BEES, was not included in the primary study samples, an additional university student sample (*n* = 326) is included in the online Appendix A. In this sample, a Pearson correlation of 0.86 was observed between the BEES and SPANE, providing further evidence of convergent validity.

After completing each questionnaire, university student participants rated the level of discomfort they experienced. The majority reported no discomfort (ranging from 53% to 76% across all measures), while only a minority reported moderate to high discomfort (4% to 15%). These findings align with previous research suggesting that completing emotional well-being questionnaires has minimal impact on respondents ([12]; [21]; [28]; [49]; [54]; [59]).

Among all measures, the BEES was associated with the lowest reported discomfort. Coupled with its brevity (median completion time of 26.4 s), this suggest the BEES may be particularly well suited for frequent or repeated use, such as in mood monitoring applications ([44]). However, we acknowledge that the BEES was also the shortest measure in the study. Further research is needed to better understand the relationship between questionnaire length and discomfort when completing emotional well-being self-report measures. For example, by investigating self-reported discomfort using the BEES alongside a range of both similarly brief tools and lengthier instruments.

We also compared BEES scores with K10 distress categories to generate preliminary cut-off values for a categorical scoring scheme. The proportion of female participants classified as experiencing high distress in each sample aligned closely with existing literature. Specifically, 26% of female university students in our sample were classified in the high distress category, consistent with prior estimates of psychological distress among Australian tertiary students ([5]; [29]; [52]). Similarly, 22% of Australian general public females were classified as high distress, matching national survey estimates of emotional well-being ([2]; [16]; [25]). Among female school students aged 11–16 years, 18% were classified in the high distress category, which is also consistent with findings from Australian ([25]; [33]; [34]; [51]), U.K. ([36]; [56]), and U.S. studies ([55]).

ROC analysis confirmed the discriminative ability of the BEES for identifying elevated psychological distress (as defined by the K10), with excellent classification accuracy (AUC = 0.89). The selected threshold of BEES < −1 provided high specificity (87.2%) and acceptable sensitivity (71.6%), making it particularly suitable for screening contexts where limiting false positives is crucial, such as schools or workplaces with limited referral resources ([18]; [20]; [60]).

The ROC analysis and convergence with previously reported prevalence rates provides preliminary support for the appropriateness of the proposed BEES cut-off values. However, further research is needed to confirm or refine these thresholds. Until more robust validation is available, we recommend that researchers and practitioners use the cut-offs proposed in this study with caution.

When interpreting BEES classification scores in the present study, while the overall proportion of participants across low, moderate and high likelihood of emotional distress was similar across samples, there was a clear gender difference with females more likely to report high emotional distress that is consistent with prior research ([11]; [16]; [53]). These findings support the scale’s robustness across diverse populations, while also highlighting the need for context-sensitive interpretation and future subgroup validation.

This study has several limitations. Although the sample sizes were large, the data were drawn primarily from Australian participants, with only one school sample from the United Kingdom. In addition, the university student sample was comprised predominately of female psychology students, which may limit the generalisability of findings to more diverse populations. Additionally, the assessment of convergent validity and self-reported discomfort was limited to the university student sample. Future research is needed with a wider variety of samples to further accumulate evidence for the reliability and validity of the BEES as a measure of emotional well-being to test for consistency across different samples and measurement contexts. Further cross-cultural research is needed to assess the broader applicability of the BEES.

Another limitation is the use of exclusively cross-sectional data. While the BEES appears suitable for a range of contexts, including repeated or momentary assessment, this flexibility assumes that its psychometric properties remain stable across different timeframes and preambles (e.g., “right now”, “past few hours”, or “this day”). Future studies should evaluate the reliability and validity of the BEES when adapted for these varied use cases. In the present study, the two adult samples reflected on the past month, while the school student sample reflected on the past week. The reliability of the BEES was consistent across these two reflection periods, but we acknowledge that the samples were different. Future research investigating the reliability of the BEES for different reflection periods across a wide range of samples is warranted.

A further limitation is the imbalance in the measures used to assess convergent validity, which were mostly distress-based (e.g., DASS-21, K10, PANAS-Negative). With only one positively framed measure included (PANAS-Positive), were unable to meaningfully compare the strength of associations across positive versus negative well-being constructs. Future studies should include a wider range of positively framed measures to better assess the specificity of the BEE’s convergent validity.

## 5. Conclusions

This study provides new evidence supporting the reliability and validity of the Brief Emotional Experience Scale (BEES) as a concise and flexible measure of emotional well-being. Compared to longer, more complex measures of emotional well-being (e.g., DASS-21, PANAS, SPANE), the BEES offers a unique combination of brevity, valence balance, and accessible language. We suggest that such properties make the BEES especially advantageous in settings where time, cognitive load, or literacy levels may be constraints. Its structure also allows flexible use in both general well-being tracking and context-specific emotional assessment. These features make the BEES particularly well-suited for use with children, in applied settings where time is limited, and in research contexts where survey space is constrained. Its short completion time and low participant discomfort further support its utility for frequent mood monitoring ([44]).

Future research is encouraged to build on the present findings by evaluating the BEES in broader cultural contexts, evaluating test-retest reliability, and examining its sensitivity to change over time and across interventions. The BEES is freely available for use without the need for formal permission. Researchers or practitioners interested in using the scale are welcome to contact the authors with any questions or to share how the tool is being applied.

## Figures and Tables

**Figure 1 behavsci-15-00643-f001:**
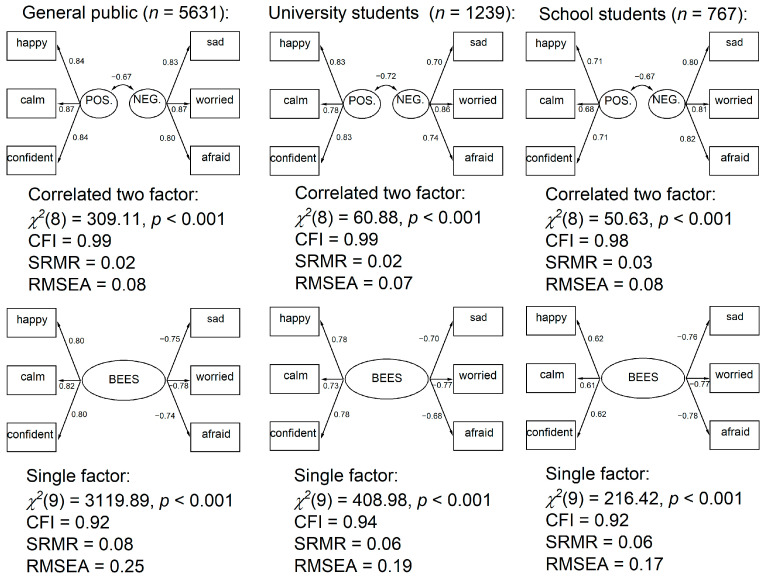
Confirmatory factor analyses across all three samples testing a correlated two factor model and single factor model.

**Figure 2 behavsci-15-00643-f002:**
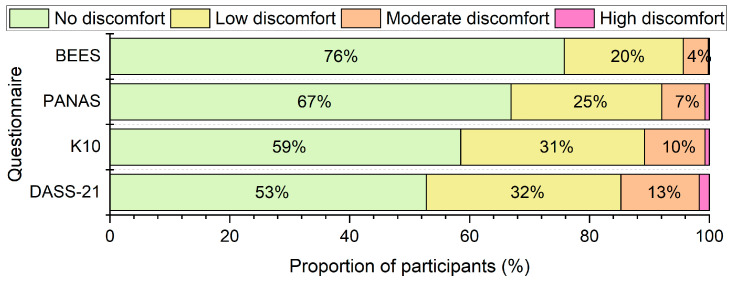
Participant self-rated discomfort when answering each of the questionnaires.

**Figure 3 behavsci-15-00643-f003:**
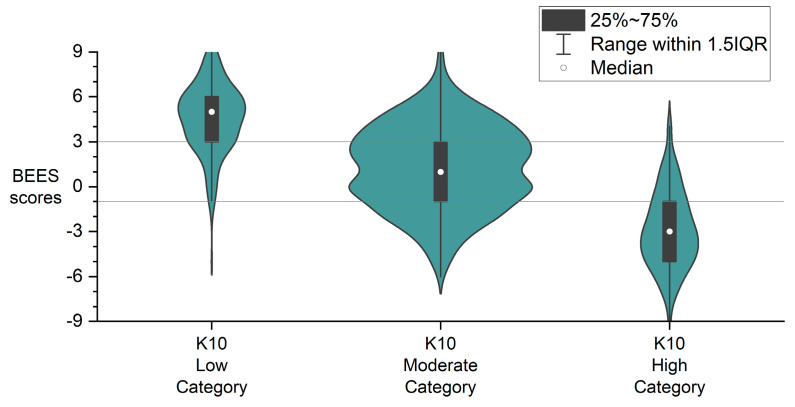
Distribution of BEES scores across K10 psychological distress categories, showing clear interquartile separation that informed proposed BEES cut-off values.

**Figure 4 behavsci-15-00643-f004:**
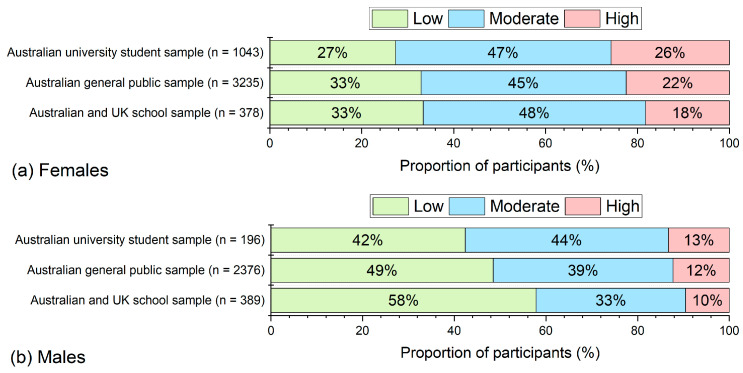
Proportion of participants in each BEES distress category (low, moderate, high) across university, general public, and school student samples.

**Figure 5 behavsci-15-00643-f005:**
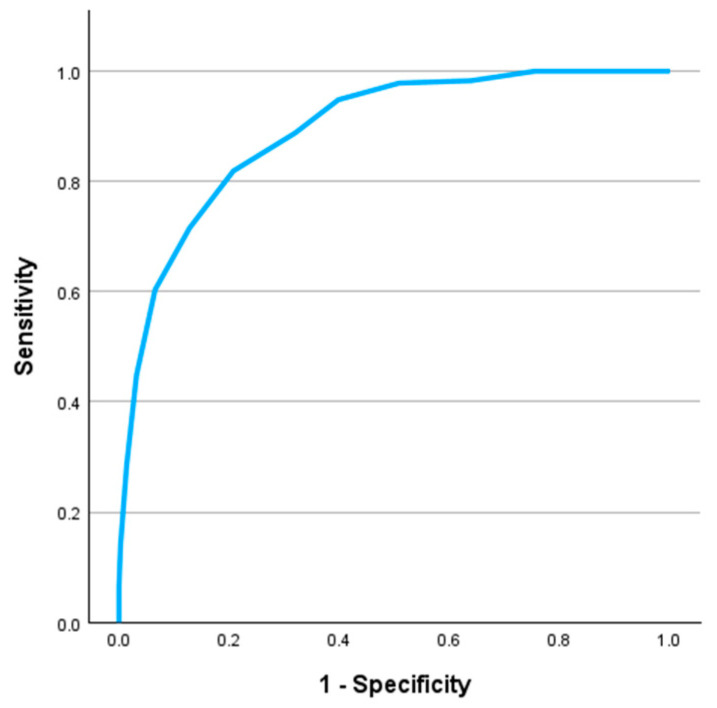
Receiver Operating Characteristic (ROC) curve for the BEES total score predicting high psychological distress (K10 ≥ 31) in the university student sample. The curve illustrates sensitivity plotted against 1 – specificity at varying cut-off thresholds.

**Table 1 behavsci-15-00643-t001:** Spearman correlations among individual BEES items across all three samples. * *p* < 0.001.

	Happy	Worried	Calm	Sad	Confident	Afraid
General public (*n* = 5631)						
Happy	-					
Worried	−0.42 *	-				
Calm	0.64 *	−0.47 *	-			
Sad	−0.45 *	0.59 *	−0.44 *	-		
Confident	0.64 *	−0.43 *	0.66 *	−0.39 *	-	
Afraid	−0.32 *	0.60 *	−0.39 *	0.56 *	−0.33 *	-
University students (*n* = 1239)						
Happy	-					
Worried	−0.40 *	-				
Calm	0.56 *	−0.48 *	-			
Sad	−0.42 *	0.54 *	−0.35 *	-		
Confident	0.60 *	−0.48 *	0.54 *	−0.43 *	-	
Afraid	−0.31 *	0.56 *	−0.35 *	0.46 *	−0.36 *	-
School students (*n* = 767)						
Happy	-					
Worried	−0.29 *	-				
Calm	0.38 *	−0.33 *	-			
Sad	−0.35 *	0.51 *	−0.37 *	-		
Confident	0.46 *	−0.30 *	0.41 *	−0.34 *	-	
Afraid	−0.28 *	0.58 *	−0.30 *	0.47 *	−0.25 *	-

**Table 2 behavsci-15-00643-t002:** Means and standard deviations () for the BEES across three samples. McDonald’s omega reliability coefficients are provided in [].

Measure	General Public		University Students		School Students	
	Females (*n* = 3235)	Males (*n* = 2376)	Females (*n* = 1043)	Males (*n* = 196)	Females (*n* = 378)	Males (*n* = 389)
BEES total	1.54 (3.78) [0.85]	2.95 (3.69) [0.81]	0.99 (3.58) [0.83]	2.44 (3.44) [0.80]	1.80 (3.57) [0.76]	3.32 (3.57) [0.79]
BEES pos.	4.65 (2.14) [0.84]	5.43 (2.20) [0.84]	4.67 (1.98) [0.80]	5.29 (1.96) [0.79]	4.59 (1.94) [0.68]	5.29 (2.04) [0.69]
BEES neg.	3.11 (2.12) [0.80]	2.48 (2.12) [0.88]	3.68 (2.03) [0.77]	2.84 (1.99) [0.73]	2.78 (2.26) [0.76]	1.97 (2.10) [0.80]

**Table 3 behavsci-15-00643-t003:** Means and standard deviations () for the emotional well-being measures filled out by the university student sample (*n* = 1239).

Measure	Female Mean (SD)	Male Mean (SD)	Median Time to Complete (Secs)
BEES	0.99 (3.54)	2.44 (3.44)	26.4
BEES (positive)	4.67 (1.98)	5.29 (1.96)	13.2
BEES (negative)	3.68 (2.03)	2.84 (1.99)	13.2
DASS-21	18.25 (11.90)	15.10 (10.18)	98.8
K10	23.07 (8.19)	21.32 (7.76)	55.3
PANAS (positive)	30.73 (7.93)	33.50 (8.48)	35.7
PANAS (negative)	22.39 (7.95)	20.38 (7.81)	35.7

**Table 4 behavsci-15-00643-t004:** Pearson correlations between the BEES and other measures of emotional well-being.

Measure	BEES	BEES (pos)	BEES (neg)	DASS-21	K10	PANAS (pos)	PANAS (neg)
BEES	-						
BEES (pos)	0.88 *	-					
BEES (neg)	−0.89 *	−0.56 *	-				
DASS-21	−0.76 *	−0.64 *	0.70 *	-			
K10	−0.75 *	−0.62 *	0.71 *	0.87 *	-		
PANAS (pos)	0.64 *	0.70 *	−0.44 *	−0.49 *	−0.51 *	-	
PANAS (neg)	−0.77 *	−0.57 *	0.79 *	0.78 *	0.76 *	−0.38 *	-

* *p* < 0.05.

**Table 5 behavsci-15-00643-t005:** Diagnostic accuracy indices for different BEES total score cut-offs in predicting high psychological distress. Sensitivity, specificity, precision (positive prediction value), and Youden’s Index are reported for each threshold.

BEES Cut-Off	Sensitivity	Specificity	Precision	Youden’s Index
<−3	44.8%	96.8%	76.5%	0.42
<−2	60.3%	93.4%	68.0%	0.54
<−1	71.6%	87.2%	56.3%	0.59
<0	81.9%	79.2%	47.6%	0.61
<1	88.8%	68.0%	39.0%	0.57

## Data Availability

Raw data for this study can be found on Figshare: https://doi.org/10.6084/m9.figshare.22237519.v7.

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
