# Peer review of "Reliability and Validity of the Brief Emotional Experience Scale (BEES) as a Measure of Emotional Well-Being"

_behavsci, 2025, doi:10.3390/bs15050643_

Round 1
Reviewer 1 Report
Comments and Suggestions for Authors
I appreciate the opportunity and invitation to review the manuscript entitled [Reliability and validity of the Brief Emotional Experience Scale (BEES) as a measure of emotional well-being]. This is a very interesting study, and the manuscript presents several significant strengths that deserve to be highlighted.
The proposal of a concise and reliable instrument for assessing emotional experience represents a valuable contribution to the field of psychological measurement. To evaluate the psychometric properties of the BEES, the authors drew on three large and distinct samples – Australian university students, the general Australian public, and school students from both Australia and the UK.
Confirmatory factor analyses (CFAs) were conducted in each of the samples, alongside convergent validity analyses using well-established wellbeing measures such as the DASS-21, PANAS, and K10. Additionally, the authors examined participant-reported discomfort when completing the questionnaire, as well as the time taken to complete the BEES compared to other measures in the study. The development of preliminary categorical cut-off scores for high emotional distress, based on K10 categories, is also a valuable contribution to the interpretation of BEES2 scores.
The BEES scale is grounded in a clear theoretical rationale inspired by the SPANE and DASS, and the deliberate selection of adjectives linked to broader constructs of emotional wellbeing is a commendable aspect of its development. Overall, the authors have conducted a rigorous study using appropriate statistical analyses, which are correctly interpreted and lead to well-supported conclusions.
Suggestions for Improvement
The following suggestions are maded intended to further enhance the manuscript’s clarity and impact and do not detract from its overall quality.
Introduction
While the introduction provides essential information on the BEES and the research context, it could benefit from improved clarity, specificity, and conciseness to more effectively engage the reader from the outset. Specifically, the following suggestions are offered:
- Specify previous findings on the BEES early in the introduction, including concrete wellbeing indicators (e.g., physical wellbeing, sense of belonging, quality of social relationships, body/weight concerns).
- Present the distinction between emotional wellbeing and emotional experience more succinctly, perhaps with a single sentence to prepare the reader for the definitions and examples that follow.
- Condense the background on existing measures (e.g., DASS-21, PANAS, SPANE) by focusing on the most relevant aspects for framing the BEES (e.g., adjective-based vs. statement-based measures, PANAS limitations regarding valence, SPANE’s influence on BEES design).
- Deepen the discussion of gaps in the literature, better articulating how this study addresses previous limitations and extends current knowledge.
- Highlight the novelty of the BEES design earlier, especially the use of deliberately paired positive and negative adjectives for structural balance.
- Ensure smoother transitions between background and study aims, using explicit linking sentences to guide the reader through the logic of the argument. For example, after presenting limitations of existing measures, more directly connect how BEES seeks to fill these gaps, leading naturally to the specific aims of the present study.
Results
The manuscript presents a robust analysis of the BEES’s reliability and validity. To enhance understanding of the scale’s properties and potential, the following complementary analyses are suggested:
- Conduct multigroup CFA to test model invariance across sex and age groups, strengthening evidence for the structural stability of the BEES.
- Explore differences in correlations with DASS-21, K10, and PANAS between men and women to assess whether convergent validity varies across demographic groups.
- Examine the magnitude of differences between correlations with distress-based (e.g., DASS-21, K10, PANAS-negative) and wellbeing-based (e.g., PANAS-positive) measures to better assess the specificity of BEES's convergent validity.
- Validate the proposed BEES cut-off scores using Receiver Operating Characteristic (ROC) analyses to improve their ability to discriminate levels of psychological distress, as measured by the K10.
Discussion
The discussion is clear and addresses the key findings of the study. However, the following enhancements are suggested to provide a deeper and more insightful interpretation:
- Relate findings more explicitly to the literature, for instance by revisiting the conceptual confusion around the PANAS and the SPANE’s intent to focus on valence, then discussing how the BEES’s two-factor structure aligns with or diverges from these theoretical bases.
- Interpret the factor structure more thoroughly, particularly the strong negative correlation between positive and negative emotion factors. This could be tied to the conceptualization of emotional wellbeing as a balance of affect, further justifying the potential use of a single overall BEES score.
- Discuss the magnitude of convergent validity coefficients (as reported in Table 3) in light of typical correlations found between emotional wellbeing measures in the literature.
- Expand on the discussion of discomfort findings, reflecting on the practical implications of lower reported discomfort with the BEES, especially in contexts of frequent or repeated mood monitoring. While BEES’s brevity is noted, the authors could more specifically suggest how future research might explore the relationship between questionnaire length and perceived discomfort (e.g., comparing BEES with similarly brief tools).
- Provide more specific future research directions, such as cross-cultural validation beyond UK student samples, test-retest reliability assessments, or sensitivity to change across time and contexts.
- Expand on study limitations, linking them more directly to interpretation and generalizability of the findings. For instance, the predominance of female psychology students in the university sample may influence both convergent validity outcomes and reported discomfort. The differing recall periods between samples (1 month vs. 1 week) may also affect comparability across age groups and populations. The cross-sectional design should also be acknowledged as a limitation regarding the stability of findings.
Author Response
Dear Reviewer 1,
Thank you for taking the time to review our manuscript. Thank you also for your kind words regarding the quality of our work. Below we give a point-by-point account of how we have made modifications in response to your suggestions for improvement.
Introduction.
Comment I1: Specify previous findings on the BEES early in the introduction, including concrete wellbeing indicators (e.g., physical wellbeing, sense of belonging, quality of social relationships, body/weight concerns).
Response I1: We now explicitly summarize previous findings on the BEES, including its associations with physical well-being, sense of belonging, social relationship quality, and body/weight concerns, in the opening paragraph (lines 29-33). This clearly contextualizes the relevance and initial utility of the measure.
Comment I2: Present the distinction between emotional wellbeing and emotional experience more succinctly, perhaps with a single sentence to prepare the reader for the definitions and examples that follow.
Response I2: We have streamlined the distinction between emotional well-being and emotional experience into a concise and clear statement, directly preceding the more detailed definitions. The distinction is now briefly introduced before elaborating with illustrative examples (lines 40-50).
Comment I3: Condense the background on existing measures (e.g., DASS-21, PANAS, SPANE) by focusing on the most relevant aspects for framing the BEES (e.g., adjective-based vs. statement-based measures, PANAS limitations regarding valence, SPANE’s influence on BEES design).
Response I3: The discussion of existing measures (DASS-21, PANAS, SPANE) has been significantly condensed. We retained only the most critical information relevant to framing the BEES, focusing explicitly on their limitations (PANAS’s confusion regarding valence/arousal) and influence (SPANE’s clear valence focus inspiring the BEES design). This is in addition to combing the overall introduction to trim it down to be more focused, down from 2328 words to 938 words. We thank the reviewer for flagging this to help us make our paper more reader friendly.
Comment I4: Deepen the discussion of gaps in the literature, better articulating how this study addresses previous limitations and extends current knowledge.
Response I4: We have expanded and clarified the articulation of specific gaps, highlighting limited prior psychometric evaluation of the BEES. We explicitly state how this study addresses these gaps, including factor structure exploration, convergent validity assessment, and categorical scoring band development, thereby clearly demonstrating the study’s contribution. By trimming down the peripheral information from the prior version, we think in this revised version the leaner introduction allows for the communication of the contribution of the study to better stand out. Explicit statements communicating the contribution of the research are embedded throughout the introduction (lines 33-39; 68-69; 78-84; 85-110).
Comment I5: Highlight the novelty of the BEES design earlier, especially the use of deliberately paired positive and negative adjectives for structural balance.
Response I5: Thank you for flagging this. In our revision of the introduction, we now have a dedicated paragraph on this point to make this point stand out better within the narrative and be clearer to the reader (lines 68-77).
Comment I6: Ensure smoother transitions between background and study aims, using explicit linking sentences to guide the reader through the logic of the argument. For example, after presenting limitations of existing measures, more directly connect how BEES seeks to fill these gaps, leading naturally to the specific aims of the present study.
Response I6: BY trimming the peripheral information from the prior version, we now feel like we do a better job having smoother transitions throughout the introduction narrative to keep the narrative more focused on the BEES and the contribution of the work. In addition to making adjustments throughout the entire introduction, we also took on board your specific suggestion to be more explicit on the transition between background and study aims by connecting the BEES more to filling the gaps from prior questionnaires to lead in better to the specific aims of the study (lines 68-110).
Results
Comment R1: Conduct multigroup CFA to test model invariance across sex and age groups, strengthening evidence for the structural stability of the BEES.
Response R1: We did as the reviewer suggested and re-ran CFA across all three samples breaking down into sex and gender groups. As the results were all consistent with the main analyses, we have decided to provide all the tables into the online supplemental files, that we refer the reader to within the manuscript on lines 206-210.
Comment R2: Explore differences in correlations with DASS-21, K10, and PANAS between men and women to assess whether convergent validity varies across demographic groups.
Response R2: We did as the reviewer suggested and ran these correlations between men and women. As these correlations did not differ from the main set of correlations, we decided to include them in the online supplemental materials. We refer the reader to that within the manuscript on lines 238-240.
Comment R3: Examine the magnitude of differences between correlations with distress-based (e.g., DASS-21, K10, PANAS-negative) and wellbeing-based (e.g., PANAS-positive) measures to better assess the specificity of BEES's convergent validity.
Response R3: As we only had one wellbeing-based versus multiple distress-based well-being measures, we did not feel like our study design could adequately incorporate a full exploration of this issue. So instead of adding material to the results section, we flagged this as a limitation with encouragement for future research to investigate this issue properly, lines 412-417. We thank the reviewer for this insightful suggestion, and we are optimistic that more work will be done in the future to further compare the BEES with a wide range of other measures.
Comment R4: Validate the proposed BEES cut-off scores using Receiver Operating Characteristic (ROC) analyses to improve their ability to discriminate levels of psychological distress, as measured by the K10.
Response R4: We have included this suggested analysis into the results section within the manuscript lines 293-307. We also have added in conclusions based on this analysis into the discussion section lines 374-384. Thank you for this suggestion. We feel by incorporating this into the paper it strengthens our evidence for the suggested cut-off values for the BEES.
Discussion
Comment D1: Relate findings more explicitly to the literature, for instance by revisiting the conceptual confusion around the PANAS and the SPANE’s intent to focus on valence, then discussing how the BEES’s two-factor structure aligns with or diverges from these theoretical bases.
Response D1: Thank you for this suggestion, we have revised the discussion to add in an additional paragraph to address this by relating the findings more explicitly to the literature. We have followed the reviewer suggestions of doing this by revisiting the conceptual confusion surrounding the PANAS and discussing how the BEES aligns more with frameworks that conceptualise emotional well-being in terms of valence balance, lines 327-338.
Comment D2: Interpret the factor structure more thoroughly, particularly the strong negative correlation between positive and negative emotion factors. This could be tied to the conceptualization of emotional wellbeing as a balance of affect, further justifying the potential use of a single overall BEES score.
Response D2: We have edited the second paragraph of discussion section to account for this reviewer suggestion to be more explicit on how the strong inverse correlation between positive and negative factors justifies the potential scoring of the BEES balanced score, lines 318-326.
Comment D3: Discuss the magnitude of convergent validity coefficients (as reported in Table 3) in light of typical correlations found between emotional wellbeing measures in the literature.
Response D3: We have included an explicit statement on the point raised by the reviewer linking the findings to what is typically found in the literature, on lines 340-342.
Comments D4: Expand on the discussion of discomfort findings, reflecting on the practical implications of lower reported discomfort with the BEES, especially in contexts of frequent or repeated mood monitoring. While BEES’s brevity is noted, the authors could more specifically suggest how future research might explore the relationship between questionnaire length and perceived discomfort (e.g., comparing BEES with similarly brief tools).
Response D4: We have taken on board the reviewer's suggestion by adding an explicit 'For example' statement being more directive on how future research could be undertaken to further investigate discomfort associated with answering emotional well-being measures, lines 360-361.
Comment D5 & D6: (5) Provide more specific future research directions, such as cross-cultural validation beyond UK student samples, test-retest reliability assessments, or sensitivity to change across time and contexts. (6) Expand on study limitations, linking them more directly to interpretation and generalizability of the findings. For instance, the predominance of female psychology students in the university sample may influence both convergent validity outcomes and reported discomfort. The differing recall periods between samples (1 month vs. 1 week) may also affect comparability across age groups and populations. The cross-sectional design should also be acknowledged as a limitation regarding the stability of findings.
Response D4 & D6: We have fleshed out the limitations and future research discussion more by including statements as per the reviewer suggestions. We have increased this part of the discussion from original 130 words to now be 313 words, lines 392-417.
Reviewer 2 Report
Comments and Suggestions for Authors
Excellent work on the presentation, statistical analysis, and sample size.
The theoretical underpinnings of emotional well-being are rather lacking. Other theories and measures of emotional well-being exist. The rationale behind using PANAS for convergent validity is unclear.
Convergent validity and factorial are explained in length, however reliability is not covered in enough information, even though the title started with reliability.
It is unclear from the discussion what your scale's advantages are. Why would a researcher attempt to use this scale among others?

Author Response
Dear Reviewer 2,
Thank you for taking the time to review our work, your kind words on the quality of our work, and your thoughtful suggestions on how we can make some modifications to further enhance the paper. We provide a point-by-point response to your feedback below.
Comment 1: The theoretical underpinnings of emotional well-being are rather lacking. Why it is crucial to
measure emotional well-being? The other current scales on emotional well-being? And the
place of BEES? It is possible to reorganize the introduction appropriately.
Response 1: We now include a general statement on the importance of measuring emotional well-being in psychological research (lines 40-42). We have also chopped down the introduction narrative of peripheral information considerably to better focus on relevant information. We now provide a more focused overview (lines 52-67) of key existing measures (e.g., DASS-21, PANAS, SPANE), explaining their strengths and weaknesses in relation to measurement format and describing the rationale for the BEES more clearly outlining its advantages in brevity, accessibility and conceptual design (lines 68-77).
Comment 2: The rationale behind using PANAS for convergent validity is unclear. Other measures of
emotional well-being exist. Why PANAS?
Response 2: Within the introduction we have clarified our rationale for including the PANAS. Specifically, we note that the PANAS was what came first in terms of historical development of adjective-based well-being measures. We also note that although it was originally developed to capture high-arousal affective states, it remains one of the most widely used adjective-based emotional self-report tools in psychological research. Its inclusion allows for comparisons with a well-established benchmark and additionally its structural contrast with the BEES (which focuses more directly on valence) provides an opportunity to evaluate conceptual overlap and divergence. Within the introduction we are now more succinct on the PANAS historical significance (lines 55-61), and provide more explicit statements on a rationale for its inclusion in the study (lines 91-96).
Comment 3: With comprehensive tables for every sample, convergent and factorial validity are thoroughly
described. One possible addition would be a table that displays the factor load for every item.
Although reliability was mentioned in the headline, there is not enough information about items.
It is possible to provide tables that illustrate inter-items correlations, Cronbach alpha if item
deleted etc.
Response 3: Thank you for flagging this with us. We have now added in a table to the results section (TABLE 1) that provides the raw inter-correlations among individual BEES items across all samples. The factor loadings for individual items are also all provided within FIGURE 1.
Comment 4: The discussion began effectively, with a clear presentation of the earlier studies that supported
the findings. A better discussion of sample-to-sample differences in outcomes would be
beneficial.
Response 4: We have added in a short paragraph on this issue flagged by the reviewer on lines 385-391.
Comment 5: BEES scale's benefits are not evident from the debate. Among various scales, why
would a researcher try to use this one? The goal and results of the conclusion section can be reorganized more clearly. The discussion may conclude with further research.
Response 5: We have doubled the length of discussion of limitations and future research by leveraging off reviewer 1 specific suggestions on some further points to include, lines 392-417. We have also reworked our final conclusion section after reflecting on reviewer 2's comments here, lines 419-435. We have taken what we feel is a stronger stance on arguing for the contribution of the BEES within the emotional well-being measurement landscape. We feel this has strengthened the end of the paper, which should increase the impact of the work, and so we thank the reviewer for bringing this to our attention.